# Sex Differences between Neuronal Loss and the Early Onset of Amyloid Deposits and Behavioral Consequences in 5xFAD Transgenic Mouse as a Model for Alzheimer’s Disease

**DOI:** 10.3390/cells12050780

**Published:** 2023-03-01

**Authors:** Chi Him Poon, San Tung Nicholas Wong, Jaydeep Roy, Yingyi Wang, Hui Wang Hujo Chan, Harry Steinbusch, Arjan Blokland, Yasin Temel, Luca Aquili, Lee Wei Lim

**Affiliations:** 1Neuromodulation Laboratory, School of Biomedical Sciences, Li Ka Shing Faculty of Medicine, The University of Hong Kong, Hong Kong SAR, China; 2Department of Neuroscience, Faculty of Health, Medicine and Life Sciences, Maastricht University, 6211 LK Maastricht, The Netherlands; 3Department of Brain & Cognitive Sciences, Daegu Gyeongbuk Institute Science and Technology (DGIST), Daegu 42988, Republic of Korea; 4Department of Neuropsychology and Psychopharmacology, Faculty of Psychology and Neuroscience, Maastricht University, 6211 LK Maastricht, The Netherlands; 5Department of Neurosurgery, Maastricht University Medical Centre, Maastricht University, 6211 LK Maastricht, The Netherlands; 6College of Health and Education, Discipline of Psychology, Murdoch University, Perth 6150, Australia

**Keywords:** Alzheimer’s disease, cognition, amyloid plaque, neuronal loss, transgenic mouse models

## Abstract

A promising direction in the research on Alzheimer’s Disease (AD) is the identification of biomarkers that better inform the disease progression of AD. However, the performance of amyloid-based biomarkers in predicting cognitive performance has been shown to be suboptimal. We hypothesise that neuronal loss could better inform cognitive impairment. We have utilised the 5xFAD transgenic mouse model that displays AD pathology at an early phase, already fully manifested after 6 months. We have evaluated the relationships between cognitive impairment, amyloid deposition, and neuronal loss in the hippocampus in both male and female mice. We observed the onset of disease characterized by the emergence of cognitive impairment in 6-month-old 5xFAD mice coinciding with the emergence of neuronal loss in the subiculum, but not amyloid pathology. We also showed that female mice exhibited significantly increased amyloid deposition in the hippocampus and entorhinal cortex, highlighting sex-related differences in the amyloid pathology of this model. Therefore, parameters based on neuronal loss might more accurately reflect disease onset and progression compared to amyloid-based biomarkers in AD patients. Moreover, sex-related differences should be considered in studies involving 5xFAD mouse models.

## 1. Introduction

It has been suggested that early pharmacological treatment during the prodromal phase of Alzheimer’s Disease (AD) might prevent amyloid from reaching levels that trigger downstream pathologies, preventing disease development and slowing down the progression in clinical settings [1]. However, prodromal AD is a period where the accumulation of amyloid occurs with little or no cognitive deficits [2], making it difficult to clinically diagnose. Therefore, a promising direction in the research on AD is to identify biomarkers that effectively correlate with the symptomatic onset of AD. Given amyloid hypothesis’ central role in the development of AD [3], several biomarkers have been developed to detect changes in amyloid levels in the brain and CSF, such as amyloid tracers like Pittsburgh compound B [4]. However, some studies have cast doubt on the association between amyloid pathology and cognitive deficits. It was shown that disease severity does not correlate with amyloid plaque load [5]. Moreover, both imaging [6] and post-mortem studies [7] have shown that people with normal cognition can have an extensive cerebral amyloid deposition. This lack of correlation might explain the observations that amyloid-based biomarkers exhibit low specificity for diagnosing AD and poorly predict disease progression [4]. Therefore, a different set of biomarkers need to be identified that can better predict the pathogenesis of AD. Transgenic mouse models can provide a platform to facilitate the identification of biomarker candidates with a good correlation with cognitive performance that could eventually be applied in human AD patients. 

Five familial Alzheimer’s Disease (5xFAD) is an APP/PS1 mouse model co-expressing five familial AD gene mutations (APP—K670N/M671L [Swedish] + I716V [Florida] + V717I [London]; PS1—M146L + L286V), which have the combined effect of rapidly driving disease pathology that recapitulates the characteristics of AD [8]. The 5xFAD mouse model has been reported to encompass early-onset amyloid pathology including intraneuronal amyloid-beta (Aβ) and cognitive deficits at 1.5 months, plaque deposition in the subiculum and cortex deep layers at 2 months, spatial memory impairment at 4–5 months, and neuronal loss in the subiculum and cortical layer V at 9 months [8]. Although there are more than 100 transgenic mouse models that each capture certain aspects of AD [9], 5xFAD mice can uniquely model neuronal loss [9] and sex-specific amyloid pathology, both of which are well-established observations in AD patients [10,11]. Female 5xFAD mice have consistently higher levels of Aβ-42 at 4, 6, and 9 months, and higher plaque density at 6 months [12]. The increase in plaque density was also shown to plateau in older female mice [13]. However, our understanding of other sex-related differences in AD pathology is limited. 

The current study investigated the relationship between cognitive deficits in the 5xFAD mouse model and Aβ pathology and neuronal loss to determine which parameter can better inform behavioural manifestations. However, most characterisation studies in 5xFAD mice overlook sex-specific differences by omitting comparisons between sexes [8] or including only one sex [14], which might have significant consequences when translating the findings to humans. To address this shortcoming, the current study assessed sex-specific behavioural deficits and neuropathology in 5xFAD mice. Furthermore, in order to identify a valid parameter that can inform the emergence of cognitive deficits in a relatively early disease stage, we have selected 4-months old and 6-months old 5xFAD mice, the age range of which has been reported to exhibit spatial working memory deficits [15]. Our study involved a large sample size with strict age definitions and directly compared between male and female 5xFAD mice. Our findings showed that the onset of the disease, including cognitive deficits, neuronal loss, and amyloid deposition, was observed at 6 months in 5xFAD mice. Moreover, female 5xFAD mice exhibited more severe amyloid pathology. Although the accumulation of amyloid preceded the other two pathologies, it was less temporally associated with cognitive impairment. Based on our observations, neuronal loss, specifically in the subiculum, could better inform the onset of cognitive deficits. 

## 2. Materials and Methods

### 2.1. Subject

The study used 5xFAD double transgenic male and female mice (Jackson Laboratory, Bar Harbor, ME, USA) with three mutations on APP (Swedish K670N, M671L; Florida I716V; and London V717I) and two mutations on PS1 (M146L and L286V) under the transcriptional control of the Thy1 promoter [8]. Mice were maintained on a C57BL/6 background. Wildtype (WT) mice with the same background were included as the control. 4- and 6-month-old male and female WT and 5xFAD mice (*n* = 10 to 12 per group) were housed in groups in the Laboratory Animal Unit at the University of Hong Kong and kept under a reverse light-dark cycle (lights off from 10:00 to 22:00) with food and water available ad libitum. Behavioral tests were conducted from 11:00 to 17:00 in the dark cycle. All animal experiments were approved by the Committee on the Use of Live Animals in Teaching and Research (CULATR) at The University of Hong Kong.

### 2.2. Open Field Test

The open field test (OFT) was performed according to the previous method [16,17]. Briefly, a mouse was placed in the centre of the arena and allowed to explore freely for 5 min. The total distance travelled in the arena, which indicates the animal’s motor function and desire to explore a novel environment, was evaluated from the video recording.

### 2.3. Y-Maze Spontaneous Alternation

A mouse was placed in one of the arms of the Y maze, with the location of the start arm randomised to minimise placement bias. The mouse was given 5 min to explore freely and the sequence of arm entries was recorded. The percentage of spontaneous alternation, which indicates the animal’s spatial working memory, was defined as the consecutive entries into the three arms divided by the number of possible alternations (total arm entry minus 2) [18].

### 2.4. Y-Maze Forced Alternation

The experimental procedure was performed as previously described [19,20]. Briefly, one of the arms of the Y maze was blocked (novel arm). A mouse was then placed in the start arm and allowed to explore freely for 5 min. After a 30-min retention period, the novel arm was unblocked and the mouse was again placed in the start arm and allowed to explore freely for another 5 min. Arm entry and percentage time spent in the novel arm (compared to the time spent in all three arms) during the retrieval phase were calculated. In both experimental settings, any mice that climbed onto the walls or escaped the maze were immediately returned to the abandoned arm. Animals with insufficient exploration time were not included in the analyses.

### 2.5. Morris Water Maze Test

The experimental procedure for Morris Water Maze (MWM) was conducted as previously described [19,20,21]. Each mouse was subject to four training trials per day for 4 days. The mouse was placed at one of the four starting locations (the sequence was randomised) and allowed to navigate to a submerged platform, and the time to reach the platform was recorded. The mouse was guided to the platform if it was unable to locate the platform in 60 s and the time to reach the platform was recorded as 60 s. The mouse was permitted to stay on the platform for 15 s before being returned to the home cage. Any mouse observed to be floating (speed less than 4 cm/s) was removed from the analyses. A long-term probe was conducted at 24 h after the last training session. The animal was placed in the pool with the platform removed and allowed to explore freely for 1 min. A reversal learning session was introduced after the long-term probe. The setup and experimental procedure were the same as above and lasted for 2 days. Reversal learning was conducted with the submerged platform relocated to the quadrant opposite the initial target quadrant. A short-term probe was conducted at 90 min after the end of the reversal training to assess short-term memory.

### 2.6. Sample Collection

All animals were anaesthetised with sodium pentobarbital overdose and perfused transcardially with ice-cold Tyrode solution followed by 4% paraformaldehyde in 0.01 M PBS. The brains were dissected, post-fixed for 24 h in 4% paraformaldehyde, and cryoprotected in 15% and 30% sucrose at 4 °C. The processed brains were then serially sectioned into ten series of 40-μm coronal slices in a cryostat (CryoStar NX50, Thermo Scientific^TM^, Waltham, MA, USA) and stored at −80 °C.

### 2.7. Histology and Immunostaining

The procedure for immunohistochemistry and subsequent cresyl violet counterstaining was modified by Tsui et al. [19]. Sections were treated with 0.5% H_2_O_2_ in 0.01 M phosphate-buffered saline-0.5% Triton X-100 (PBS-T) for 15 min to inhibit endogenous peroxidase. Sections were incubated in 1% bovine serum albumin in 0.01 M PBS at room temperature for 15 min to reduce non-specific antibody binding. For antigen detection, sections were incubated in mouse anti β-amyloid (4G8) (1:500, 800,872, BioLegend, San Diego, CA, USA) mouse anti-NeuN (1:500, MAB377, Merck Millipore, Burlington, MA, USA) and rabbit anti-GFAP (1:1000, AB5804, Sigma-Aldrich, St. Louis, MO, USA) overnight at 4 °C, followed by the corresponding secondary biotinylated goat anti-mouse IgG antibody (1:500, BP-9200-50, Vector Laboratories, Newark, CA, USA) or goat anti-rabbit IgG antibody (1:500, BA-91000-50, Vector Laboratories, USA) for 1.5 h at room temperature. The antigens were visualised by adding avidin and biotinylated horse radish peroxidase (1:1000, Vectastain, Vector Laboratories, USA) in combination with 0.05% 3,3′-diaminobenzidene tetrahydrochloride, 0.15% nickel chloride, and 0.005% H_2_O_2_ in 0.05 M Tris buffer. Heparin (15 U/mL) was added to all solutions to inhibit endogenous RNAase [22]. 

Stained sections were mounted on gelatine-coated slides and dehydrated in graded ethanol solutions and then cleared in xylene. For counterstaining, the sections were rehydrated in graded ethanol and MilliQ water and counterstained with 0.1% cresyl violet solution. The glacial acetic acid in 95% ethanol was added until the desired differentiation was achieved. The sections were dehydrated, cleared, and coverslipped with Fisher Chemical Permount (Fisher Scientific, Hampton, NH, USA) for light microscopy. Sections were examined under a Zeiss Axioplan 2 epifluorescent microscope equipped with an Olympus DP73 camera (OIympus, Tokyo, Japan).

### 2.8. Quantification of the Chromogenic Staining

For the quantitative analyses, all pictures were randomised and the assessors were blinded to the exact sequences. 3–4 stained brain sections per mouse were used for analysis. For 4G8 and GFAP staining, photographs were taken to capture each area of interest (subiculum, cornu ammonis 1–3, dentate gyrus, and entorhinal cortex) in three sections of the same series using a 10× objective [19]. Each region was delineated and a threshold (half of the mean background) was applied in ImageJ (NIH, Bethesda, MD, USA). The percentage area of positive staining with an intensity above the threshold was calculated. For NeuN staining, three sections were chosen and three photographs from each area of interest were taken using a 40× objective. Positive neuronal nuclei, defined as staining with an intensity above the threshold (half of the mean background), were counted manually. The results were converted to cell/mm^2^ for analysis. 

### 2.9. Statistical Analysis

The data were first assessed by the Shapiro-Wilk test for examining the data normality distribution. The behavioral test results were analyzed by either three-way analysis of variance (ANOVA), or four-way mixed ANOVA with Bonferroni correction. One-way ANOVA was used to evaluate 4G8 plaque density in 5xFAD mice in different age groups and two-way ANOVA was used for neuronal density. Orthogonal planned comparisons were applied *a priori* to evaluate the within subgroup differences. The level of statistical significance was set at *p* < 0.05. All statistical analyses were carried out in IBM SPSS Statistics 25 and graphs were plotted with GraphPad Prism 7.0.

## 3. Results

### 3.1. 5xFAD Mice Show Onset of Hippocampus-Dependent Cognitive Deficits at the Age of 6 Months

The 5xFAD transgenic mouse model exhibits early-onset AD pathology and cognitive deficits [8]. However, the age definitions in the animal model have been rather loose in past studies, which may limit its use in determining therapeutic treatment efficacy. Previously, we suggested that age and disease stage are essential factors when examining cognitive deficits and therapeutic treatments in AD transgenic animal models [23]. In the current study, we aimed to characterize the age at which the disease manifests in the 5xFAD transgenic mouse model. We first performed an OFT to assess the locomotor activity of the animals. Generally, female 5xFAD mice showed reduced distance travelled in the open field compared to males, as indicated by three-way ANOVA revealing significant main effects of Sex (F_(1, 70)_ = 106.51, *p* < 0.001) and Genotype (F_(1, 70)_ = 6.492, *p* = 0.013) in the distance travelled in the arena. *A priori* analysis with orthogonal planned comparisons showed a significant reduction in the distance travelled in 4- and 6-month-old male 5xFAD mice compared to 4-month-old WT mice (*p* < 0.05) (Figure 1A). Although there were no observed differences between WT and 5xFAD female mice in the corresponding age groups, 6-month-old female 5xFAD mice showed hypoactivity compared to 4-month-old female 5xFAD mice (*p* < 0.05) (Figure 1B). These results demonstrate the tendency of hypoactivity in both male and female 5xFAD mice. Next, we performed Y-maze forced alternation and MWM to assess spatial memory. These two tests have been demonstrated to heavily involve hippocampal functions [24,25], and thus provide an appropriate way to characterize disease outcomes in this study. In the Y-maze forced alternation, three-way ANOVA revealed significant main effects of Sex (F_(1, 79)_ = 10.668, *p* = 0.002) and Genotype (F_(1, 79)_ = 23.795, *p* < 0.001). Both male and female 5xFAD mice exhibited spatial memory impairment at 6 months, as indicated by a significantly reduced time spent in the novel arm during the retrieval phase (*p* < 0.05). However, there was a more prominent deficit in male 5xFAD mice, with the time spent in the novel arm significantly reduced in 6-month-old male 5xFAD mice compared to their WT counterparts (*p* < 0.001), but this was not observed in female mice (Figure 1C,D). The spatial memory deficit was further confirmed by the MWM, which can examine spatial learning, and short- and long-term spatial memory. The four-way repeated measures ANOVA identified significant main effects of Time (F_(3, 243)_ = 138.98, *p* < 0.001), Genotype, and interactions of Age × Sex, Age × Genotype, Sex × Genotype (all F_(1, 81)_ > 5.34, *p* < 0.05), Time × Age, Time × Age × Sex, and Time × Age × Genotype (all F_(3, 243)_ > 3.16, *p* < 0.025), suggesting the differential ability of animals in locating the platform (Figure 1E). Male WT mice showed similar learning abilities in both age groups, as demonstrated by a comparable escape latency. A mild learning deficit was observed in 4-month-old male 5xFAD mice, as indicated by a significantly longer escape latency on day 2 of training (*p* < 0.05). Importantly, learning deficits were exacerbated in 6-month-old male 5xFAD mice, as shown by longer escape latency compared to 4-month-old 5xFAD and 6-month-old WT male mice. The 2-day reversal training following the long-term probe was designed to assess the cognitive flexibility of the mice (Figure 1E). Significant main effects of Genotype and Time × Genotype (all F_(1, 80)_ > 4.1, *p* < 0.05) were observed in the reversal learning trials. The 4- and 6-month-old male 5xFAD mice took significantly longer to locate the hidden platform compared to their respective WT counterparts on day 7, indicating impaired reversal learning. However, only a significant main effect of Age (F_(1, 68)_ = 4.11, *p* = 0.047) was identified in the long-term memory, and the multiple comparisons did not show any significant differences among groups (Figure 1G). In addition to this observation, there were significant main effects of Genotype, Age × Sex, and Age × Genotype (all F_(1, 76)_ > 7.53, *p* < 0.01) in the short-term probe. *A priori* analysis confirmed the impairment of short-term memory in 6-month-old but not 4-month-old male 5xFAD mice compared to 4-month-old male WT mice (Figure 1H). Female mice did not show any significant differences in spatial learning in 4-month-old 5xFAD or WT groups (Figure 1F). However, 6-month-old female WT mice exhibited mild learning deficits, as shown by a longer escape latency starting from training day 2. Similar to their male counterparts, 6-month-old 5xFAD female mice took a significantly longer time to locate the hidden platform during the training sessions compared to their WT counterparts, which indicates a more severe learning deficit in the transgenic mice. In contrast to male mice, there was an insignificant decrease in the escape latency observed in 6-month-old female 5xFAD mice compared to their WT counterparts. Female WT mice in both age groups demonstrated comparable long-term memory, as indicated by the similar time spent locating the target quadrant during the long-term probe. The 6-month-old female 5xFAD mice spent significantly less time in the target quadrant compared to 4-month-old female 5xFAD mice (*p* < 0.05), suggesting impairment of long-term memory (Figure 1I). No significant difference was detected in the reversal learning, suggesting comparatively intact cognitive flexibility in female 5xFAD mice. However, 6-month-old female 5xFAD mice showed impaired short-term memory, as indicated by a reduced time spent in the target quadrant in the short-term probe (Figure 1J). Overall, the above observations showed that both male and female 4-month-old 5xFAD mice have early onset mild deficits in spatial learning, but long-term memory remains relatively unaffected. Cognitive deficits became more pronounced as the animals aged, as shown by the behavioural results from Y-maze forced alternation and MWM in 6-month-old 5xFAD mice.

### 3.2. Differences in Amyloid Pathology in Male and Female 5xFAD Mice

To further explore the mechanism behind the differential behavioral performances of male and female 5xFAD mice, we examined amyloid plaque deposits in brain sections stained with 4G8 antibody, which targets amino acid residues 17–24 of β-amyloid. As cognitive impairment is a prominent feature in AD, we focused on related brain regions including the subiculum (Sub), the hippocampus, and the entorhinal cortex (ENT) (Figure 2A,B). At 4 months, amyloid plaque pathology was observed in all three examined brain regions. Plaque density was highest in the subiculum regardless of age and sex. There was a main effect of Age (all F_(1, 18)_ > 5.479, *p* < 0.05), which clearly demonstrates the progression of amyloid pathology from 4 to 6 months. *A priori* analysis showed significant amyloid progression in only the CA1–2 hippocampal region of male 5xFAD mice (*p* < 0.05), whereas there was a marked difference in all three brain regions in female 5xFAD mice (all *p* < 0.05). The rate of amyloid progression was accelerated in female 5xFAD mice and was manifest as a more severe plaque pathology in CA1–2, CA3, and DG of the hippocampus, as well as ENT, as the main effects of Sex on amyloid deposition were observed in these regions (all F_(1, 18)_ > 6.456, *p* < 0.05). *A priori* analysis confirmed that the amyloid plaque load in DG and ENT was significantly higher in 6-months-old female 5xFAD mice (DG: *p* < 0.05; ENT: *p* < 0.001). 

### 3.3. Neuronal Loss in the Subiculum of 6-Month-Old 5xFAD Mice

The 5xFAD mouse model is one of a few AD transgenic mouse models that display neuronal loss. As neuronal loss is correlated with memory impairment, we stained mature neurones with anti-NeuN antibodies to examine this pathology (Figure 2C). There were main effects of Age, Sex, Genotype, and Age × Sex (all F_(1, 32)_ > 4.87, *p* < 0.05). *A priori* analysis revealed a significant increase in the neuronal number in CA1 and CA3 of 6-month-old 5XFAD mice (Male: *p* < 0.05; Female: *p* < 0.01), whereas neuron numbers were comparable in 4-month-old 5XFAD and WT mice. Further, *A priori* analysis did not reveal any significant sex differences in neuronal number at 6 months (Figure 2D). Interestingly, neuron number in female WT mice was higher at 6 months than at 4 months (*p* < 0.001). 

An in-depth analysis of neuron density in the subiculum revealed significant effects of Age, Sex, Genotype, and Age × Sex (All F_(1,32)_ > 4.86, *p* < 0.05). A significant decrease in the number of NeuN^+^ cells was observed in both male and female 5xFAD mice when they reached 6-month old, implying the emergence of neuronal loss in 5xFAD at 6-month old (Figure 3A,B). Thus, the onset of neuronal loss in 5xFAD starts at 6 months, coinciding with the onset of behavioural impairment.

### 3.4. Increased Neuroinflammation in 5xFAD Mice at 6 Months

Astrocytic activation is a prominent hallmark of AD rodent models and patients. To characterize astrocytic activity in 5xFAD mice, brain sections were stained with a glial fibrillary acid protein (GFAP) antibody to visualize astrogliosis (Figure 4A). Given that severe amyloid pathology was observed in the hippocampus by 4G8 staining, we focused on GFAP expression changes in the hippocampus. We found no significant differences in hippocampal GFAP expression between WT and 5xFAD male mice regardless of age. However, the region-specific analysis revealed enhanced GFAP expression in the CA1 and DG of 6-month-old 5xFAD mice compared to their WT counterparts (Figure 4B), indicating increased neuroinflammation in male 5xFAD mice at 6 months. On the other hand, female 5xFAD mice displayed accelerated neuroinflammation, as indicated by an increase in GFAP^+^ cells in the hippocampus at 4 months. Among all groups, GFAP intensity was highest in the hippocampus of 6-month-old female 5xFAD mice, suggesting a more rapid disease progression in female mice compared to males (Figure 4C).

## 4. Discussion

Neuronal loss and cognitive impairment are prominent symptoms of AD, and are the main parameters for assessing the efficacy of therapeutic treatments in animal models. Various rodent models of AD that exhibit specific pathophysiology and degrees of disease severity have been developed for research. Therefore, it is crucial to identify appropriate AD animal models for downstream investigation of disease onset age and sex. Here, we provide in vivo evidence supporting strict age definitions of the onset of AD pathology in 5xFAD mice and also sex-specific differences. In terms of temporal progression, amyloid pathology manifests before 4 months, followed by hypoactivity at 4 months, and then neuronal loss and spatial memory impairment at 6 months. We also observed sex-specific differences in AD pathology, with more severe amyloid pathology and more rapid disease progression in female 5xFAD mice compared to male 5xFAD mice.

Our results strengthen the findings from previous studies conducted on 5xFAD mice, which showed the onset of plaque deposition at 2 months in both the subiculum and cortical layer V, which preceded other observed pathologies [8]. Moreover, 6-month-old male 5xFAD mice showed consistently worse performance in learning and spatial memory functions in the MWM and Y maze compared with control mice [26,27,28,29,30,31]. To the best of our knowledge, neuronal loss in the subiculum occurring at 6 months has not been previously reported. Two characterisation studies on 5xFAD mice previously reported the onset of neuronal loss after 9 months [8,14]. Such a discrepancy can be possibly explained by the different study methodologies. Oakley et al. did not perform neuronal counting prior to 9 months [8]. Hence, it is possible that neuronal loss occurred at an earlier time point, but was not detected. Moreover, the cresyl violet staining method used in both these studies is less specific than NeuN staining, which can lead to glial cells being misclassified as small neurones, thereby exaggerating the final neuronal count and obscuring subtle differences [32]. Another finding in our study is that 5xFAD mice showed hypoactivity in the OFT at 4 months, which has not been reported to occur at such an early age [15]. The underlying mechanism behind the hypoactivity has been postulated to be related to axonal transport impairment due to intraneuronal Aβ in the brain and spinal cord [15], but research is lacking regarding this aspect of AD pathology.

Mouse models are frequently used to evaluate drug efficacy in pharmacological studies. In AD, however, the inability to translate treatment success from mouse preclinical studies to human patients in clinical settings casts doubt on the appropriateness of using mice to model AD [33]. Some researchers have proposed that AD mouse models should only be used in the preclinical stages of AD, when irreversible pathologies like neuronal loss and synaptic deficits have yet to manifest, as the observed efficacy of disease-modifying treatments might be exaggerated [1]. This has been demonstrated in 5xFAD mice, in which the knockout of BACE1 was able to restore fear memory and reduce amyloid plaque load at 6 months, but showed no significant benefit at 15–18 months [34]. The prudent use of older mice in pharmacological studies should be advocated to avoid exaggerating treatment efficacy. Another avenue that shows promise for the treatment of AD is to screen asymptomatic individuals for secondary prevention prior to the onset of irreversible pathologies [35]. To this end, 5xFAD mice might be suitable for evaluating the efficacy of early interventions for secondary prevention, but researchers have rarely considered early mouse neural development. The window for secondary prevention in 5xFAD mice is between the onset of amyloid pathology at 1.5 months and the onset of cognitive deficits at 6 months, which coincides with the growth and maturation of multiple CNS regions that continues until 11 weeks [36]. The efficacy of interventions, especially long-term ones, might be confounded by unaccounted developmental effects in mice. Other AD transgenic models with a more indolent and progressive disease course would be more suitable for this purpose.

Given that females, whether human or mouse, are more susceptible to AD and display a more severe pathology [12,37], we hypothesised that female 5xFAD mice would demonstrate a more severe pathology than male mice. In terms of amyloid pathology, we observed that female 5xFAD mice displayed accelerated disease progression. There was significant amyloid deposition in all brain regions from 4 to 6 months, whereas a significant increase of amyloid was observed in only the CA1–2 of male 5xFAD mice. In addition, amyloid deposition in the DG and ENT of 6-month-old female 5xFAD mice was significantly higher than in their male counterparts. This sex difference has been hypothesised to be due to the effects of oestrogen on the oestrogen response element in the Thy1 promoter used to drive transgene expression in 5xFAD mice, and thus is an experimental artefact rather than a true physiological difference [12]. Indeed, astrocytic activation has emerged as a key hallmark in the AD brain. Although previous studies in preclinical AD animal models generally reported the occurrence of astrocytic hyperactivation in the later disease stages (~8–9 months) [38,39], we observed neuroinflammation occurred at an earlier stage, specifically in female 5xFAD mice. Our immunohistochemical results provide strong evidence that aligns with the findings from a recent transcriptomic analysis by Bundy et al., which found upregulated genes related to the immune system and glial activation in female 5xFAD mice compared with male mice [40]. Another study showed that female 5xFAD mice expressed more markers of neuroinflammation than male mice [41], suggesting females had pronounced neuroinflammation. Hence, neuroinflammation could further drive amyloid pathology development via the secretion of inflammatory cytokines [42]. Our findings in the current study resemble the sex-dependent disease severity seen in clinical settings, research which have previously identified a number of female-biased changes that are associated with amyloid and tau pathology. For example, serpin family B, a protein family that is linked to amyloidosis, and CLDN16, a protein thought to contribute to tau pathology, were found to be positively and negatively correlated with their respective pathology exclusively in female but not male AD patients [43]. Microglia from male and female also exhibit differential responses to stimuli. Microglia phagocytosis was proposed as one of the possible amyloid clearing mechanisms in AD [44]. A recent study focusing on microglia microRNAs that modulate its phagocytosis activity revealed that gene enrichment changes involved in inflammation and phagocytosis are more pronounced in microglia from males compared to that of females [45]. These studies provide insights on sex-specific mechanisms that contribute to and/or counter AD pathology. Although the pathological mechanism behind sex differences in AD has yet to be elucidated, the results of our present study and previous reports demonstrate the viability of using 5xFAD mice as a model to investigate the causes of such differences. More in-depth studies are needed to understand this sex-biased pathology in AD.

As amyloid pathology is closely associated with neuronal loss and females tend to have more severe amyloid pathology, a logical assumption would be that the neuronal loss is exacerbated in females. Surprisingly, our study did not show any significant differences in neuronal number between male and female 5xFAD mice at either 4 or 6 months. To the best of our knowledge, sex differences in neuronal loss have not been reported because most available studies included only one sex or did not analyse male and female mice as separate subgroups. Nevertheless, there are several reasons to explain the absence of sex differences in neuronal loss: (1) although a sex difference in neuronal loss might exist, it is too small to be detected in 6-month-old mice; (2) the level of the amyloid beta oligomer, the effector of neuronal loss [46], does not show sex differences [47]; and (3) neuronal loss is independent of the amyloid level after a certain threshold has been reached [1]. More research is needed to determine whether there is a sex difference in neuronal loss in AD.

In conclusion, our study establishes the age of onset of multiple AD hallmark pathologies, including amyloid plaques, neuronal loss, learning deficits, spatial memory impairments, and hypoactivity. Through demonstrating onset age and sex differences in AD, our study highlights the need to consider the age and sex of animals when designing preclinical studies to ensure an accurate interpretation of the results.

## Figures and Tables

**Figure 1 cells-12-00780-f001:**
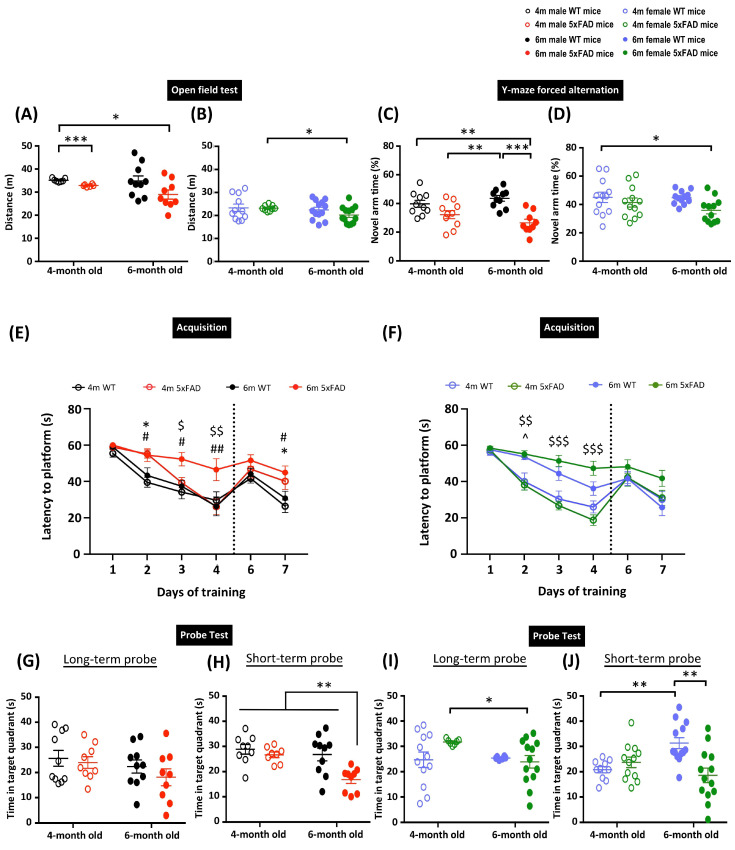
Memory deficits in 5xFAD mice at 6 months. Distance traveled in the open field test in (**A**) male and (**B**) female 5xFAD mice. Percentage of time spent in the novel arm in Y-maze forced alternation in (**C**) male and (**D**) female 5xFAD mice. Learning performance during the acquisition phase of the Morris water maze by the latency to locate the hidden platform in (**E**) male and (**F**) female 5xFAD mice. Reversal learning with the platform moved to the opposite quadrant was assessed on days 6–7. Time spent in the target quadrant during the long- and short-term probe in (**G**,**H**) male and (**I**,**J**) female 5xFAD mice. Data presented as mean ± S.E.M. Scatterplot shows individual value; * denotes comparison between 4-months old WT and 4-months old 5xFAD mice, ^ denotes comparison between 4-months old and 6-months old WT mice, $ denotes comparison between 4-months and 6-months old 5xFAD mice, # denotes comparison between 6-months old WT and 6-months old 5xFAD mice; *, ^, $, # *p* < 0.05, **, $$, ## *p* < 0.01, ***, $$$, *p* < 0.001.

**Figure 2 cells-12-00780-f002:**
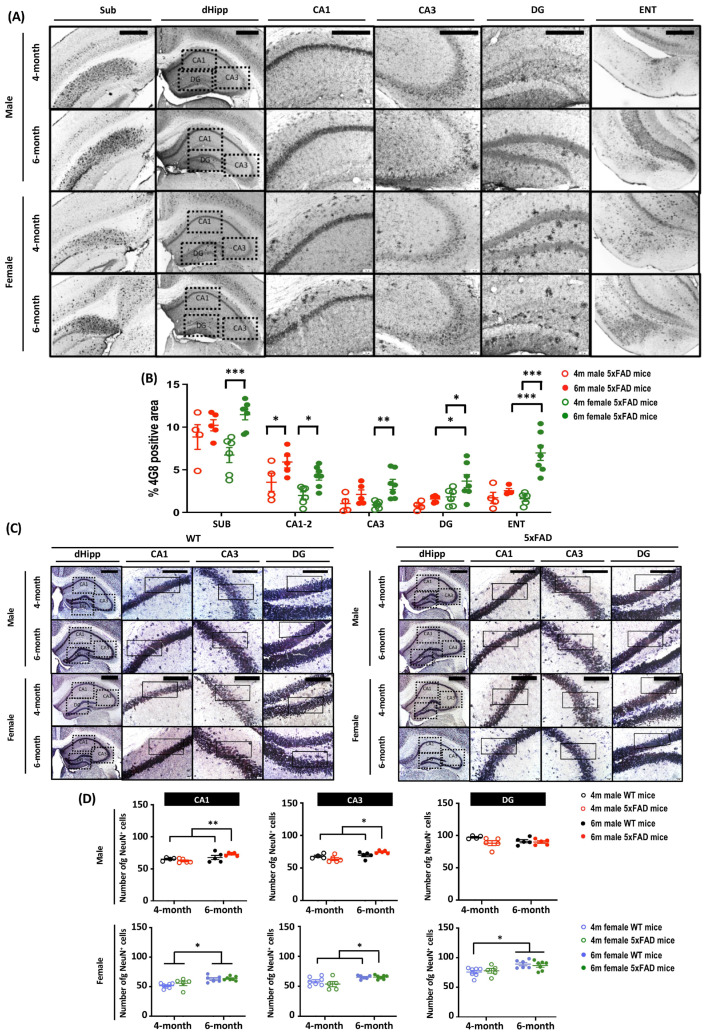
Amyloid burden and neuronal loss in brain regions related to memory in 4- and 6-month-old male and female 5xFAD mice. Diffused amyloid deposition was observed in brain regions including (**A**) subiculum (Sub), hippocampus (CA1–3 and dentate gyrus, DG), and entorhinal cortex (ENT). Scale bar: Sub, 500 μm; dHipp, CA1–3 and DG, 200 μm; ENT, 500 μm. (**B**) The main effect of Age was observed in all brain regions (Sub: F_(1, 18)_ = 12.261, *p* = 0.003; CA1–2: F_(1, 18)_ = 13.394, *p* = 0.002; CA3: F_(1, 18)_ = 11.276, *p* = 0.004; DG: F_(1, 18)_ = 5.479, *p* = 0.031; ENT: F_(1, 18)_ = 17.819, *p* = 0.001), along with main effect in Sex in CA1–2 (F_(1, 18)_ = 6.456, *p* = 0.02), DG (F_(1, 18)_ = 7.673, *p* = 0.013) and ENT (F_(1, 18)_ = 8.762, *p* = 0.009). Pairwise comparisons showed significant increases in amyloid burden in all brain regions in female 5xFAD mice (Sub: *p* < 0.001; CA1–2: *p* = 0.012; CA3: *p* = 0.002; DG: *p* = 0.021; ENT: *p* < 0.001), whereas the significant increase of amyloid was observed in only the CA1–2 (*p* = 0.025) in male 5xFAD mice, indicating a sex-dependent effects on amyloid progression. The amyloid burden was significantly higher in the DG (*p* = 0.014) and ENT (*p* < 0.001) in 6-month-old female mice. Overall, the findings indicate a possible sex-dependent effect, with a more severe amyloid pathology observed in female 5xFAD mice. (**C**) Representative images of mature neurones in hippocampal brain sections stained with anti-NeuN antibody. Scale bar: 200 μm. (**D**) For male mice, two-way ANOVA revealed significant main effects of Age in CA1–2 (F_(1,15)_ = 9.335, *p* = 0.008); main effects of Age (F_(1, 15)_ = 10.883, *p* = 0.005) and Age × Genotype (F_(1, 15)_ = 5.892, *p* = 0.028) in CA3; but no significant effect in the DG. Female mice showed a significant increase in the number of NeuN^+^ mature neurones at 6 months. Two-way ANOVA showed significant main effects of Age in CA1–2, CA3, and DG (all F_(1, 21)_ > 12.57, *p* < 0.01). Data presented as mean ± S.E.M. for comparisons. Scatterplot shows individual values. (*n* = 3–7; * *p* < 0.05, ** *p* < 0.01, *** *p* < 0.001. SUB = subiculum; CA = cornu ammonis; DG = dentate gyrus; ENT = entorhinal cortex).

**Figure 3 cells-12-00780-f003:**
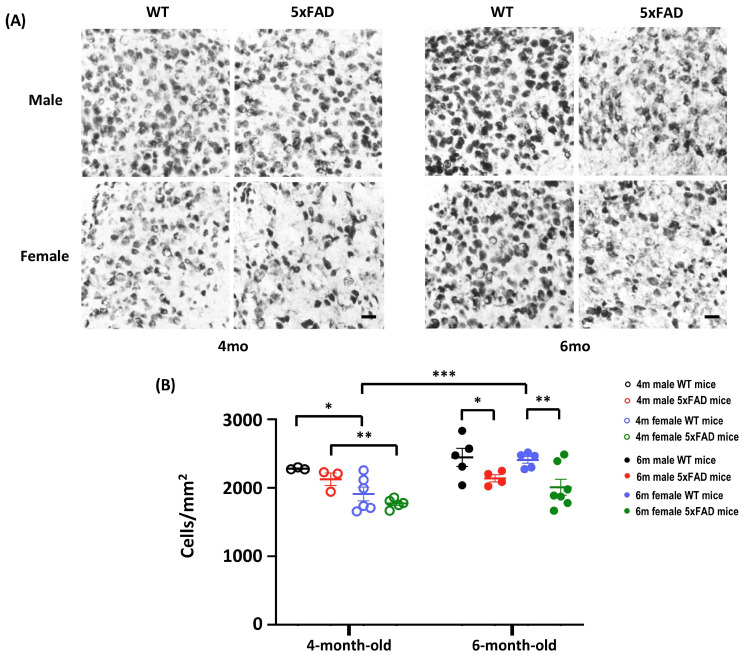
Neuronal count in the subiculum of 4- and 6-month-old WT and 5xFAD mice. (**A**) Representative images of subiculum sections stained with anti-NeuN from 4- and 6-month-old mice. A sparser neurone density was observed in 6-month-old 5xFAD mice. Scale bar: 20 μm. (**B**) The NeuN^+^ mature neurones showed main effects of Age (F_(1, 32)_ = 9.495, *p* = 0.004), Sex (F_(1, 32)_ = 12.467, *p* = 0.001), Genotype (F_(1, 32)_ = 10.713, *p* = 0.003), and Age × Sex (F_(1, 32)_ = 4.87, *p* = 0.035). *A Priori* analysis showed significant differences in the number of mature neurones in WT and 5xFAD mice at 6 months (Male: *p* = 0.039; Female: *p* = 0.003), but not at 4 months, indicating the onset of neuronal loss at 6 months. *A Priori* analysis also revealed sex-dependent differences in the number of neurones at 4 months (Male WT vs. Female WT: *p* = 0.02; Male 5xFAD vs. Female 5xFAD: *p* = 0.005), and a higher number of neurones in 6-month-old female WT mice compared to 4-month-old counterparts (*p* = 0.001). Data presented as mean ± S.E.M. for comparisons. Scatterplot shows individual values. (*n* = 3–7; * *p* < 0.05, ** *p* < 0.01, *** *p* < 0.001).

**Figure 4 cells-12-00780-f004:**
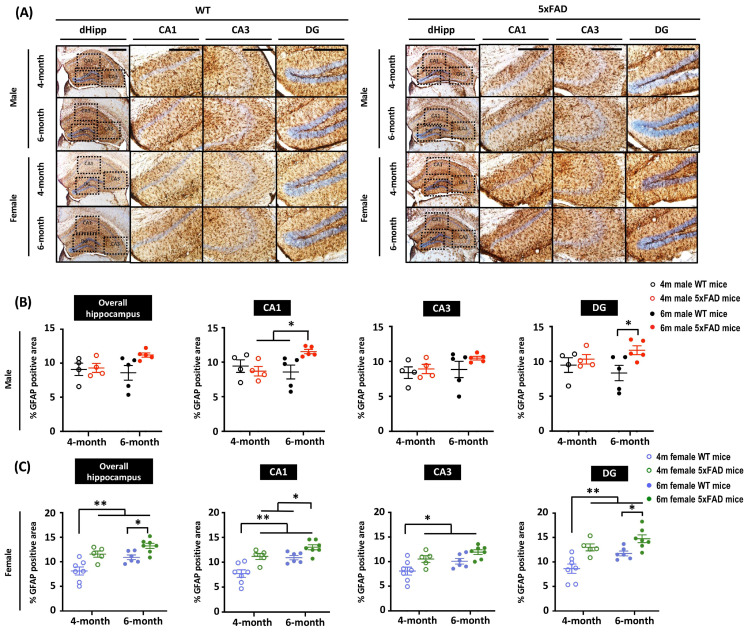
Enhanced astrogliosis was observed in 6-month-old female 5xFAD mice, but not in male 5xFAD mice. (**A**) Representative images of hippocampal sections stained with anti-GFAP antibody from 4- and 6-month-old mice. Scale bar: 200 μm. (**B**) Quantification of GFAP intensity in sections from male animals shows significant main effects of Age × Genotype (F_(1, 14)_ = 5.605, *p* = 0.033) in the CA1 and Genotype (F_(1, 14)_ = 18.901, *p* = 0.04) in the DG. Multiple comparisons showed the 6-month-old male 5xFAD mice had the highest GFAP intensity. (**C**) Two-way ANOVA showed significant main effects of Age (F_(1, 21)_ = 11.725, *p* = 0.003) and Genotype (F_(1, 21)_ = 19.475, *p* < 0.001) on GFAP intensity in the hippocampal region of female 5xFAD mice. Analysis based on sub-regions revealed significant main effects of Age (CA1: F_(1, 21)_ = 17.208, *p* < 0.001; CA3: F_(1, 21)_ = 6.681, *p* = 0.017; DG: F_(1, 21)_ = 9.353, *p* = 0.006) and Genotype (CA1: F_(1, 21)_ = 20.631, *p* < 0.001; CA3: F_(1, 21)_ = 10.645, *p* = 0.004; DG: F_(1, 21)_ = 21.288, *p* < 0.001). Multiple comparisons showed increased GFAP intensity in 4-month-old female 5xFAD mice, indicating accelerated astrocytic activation. Data presented as mean ± S.E.M. for comparisons. Scatterplot shows individual values. (*n* = 3–7; * *p* < 0.05, ** *p* < 0.01).

## Data Availability

The data presented in this study are available on request from the corresponding author.

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
