# Peer review of "Sex Differences between Neuronal Loss and the Early Onset of Amyloid Deposits and Behavioral Consequences in 5xFAD Transgenic Mouse as a Model for Alzheimer’s Disease"

_cells, 2023, doi:10.3390/cells12050780_

Round 1
Reviewer 1 Report
1. In the manuscript: Sex differences between neuronal loss and the early onset of amyloid deposits and behavioral consequences in 5xFAD transgenic mouse as a model for Alzheimer’s disease, the authors showed a significant gender influence in interesting parameters associated with neuronal loss and cognitive impairment in early stage of AD using a specific transgenic mice model with 4 or 6 age months and compared with WT. However, the authors cited in the introduction section it is known that 5xFAD mouse model present the AD related alterations in determinate ages: "The 5xFAD mouse model has been reported to encompass early-onset amyloid pathology including intraneuronal amyloid-beta (Aβ) and cognitive deficits at 1.5 months, plaque deposition in the subiculum and cortex deep layers at 2 months, spatial memory impairment at 4 – 5 months, and neuronal loss in the subiculum and cortical layer V at 9 months [8]'. On the other hand, the authors concluded that: "our study establishes the age of onset of multiple AD hallmark pathologies, including amyloid plaques, neuronal loss, learning deficits, spatial memory impairments, and hypoactivity. Through demonstrating onset age and gender differences". The question is if the amyloid plaques deposition and neuronal loss is report in the same animal model at 1.5 and 2 months, how the authors can conclude that they established the age of onset if they evaluated just 4- and 6-months age animals? The reported results could have been started early (e.g., 2 or 3 months)? In my opinion, a temporal study could confirm the information and also shows if male or female present that alterations at different ages or not. The authors should insert a shortly justificative to explain why they evaluated just 4- or 6-months age mice in the present study in introduction or discussion section.
2. Major review in Methods.
In the Quantification of the chromogenic staining, it is not clarified the quantification criteria applied. The authors need describe more details.
- "Each region was delineated, and a threshold (half of the mean background) was applied in ImageJ (NIH, USA). The percentage area of positive staining with an intensity above the threshold was calculated" Is it means the results represent the integrated density of pixels?
- The quantification criteria were the same for all antibodies.
- How many sections of each animal was quantified for each marker?
- What was the interval in μm between sections?
- What criteria was applied for select a representative area?
- The quantification was performed in both right and left side of the brain? If not, why?
- To describe neuronal loss the authors performed an immunolabelling with NeuN marker and counted the number of positive cells. The authors consider both brain side. The count was performed in a representative area each side (fill image or a specific area).? The authors used a correction formula to obtain final result. (e.g., to corrects double counting of motoneurons, because the same cell may be present in two sections, it is commonly used the Abercrombie’s formula). Could be applicable for this brain neurons? In addition, is not clear how the NeuN results were converted to cell/mm2.
- How was performed the GFAP quantification? The data is showing as a relative expression but no explanation about that was find in methods.
Minor comments:
Methods section:
- When we perform behavioral tests, it is very important detailed the time of occurred (e.g., day or night, always at the same hour or not), due the circadian cycle in rodents. Please include the complete information in the text.
- the item "Animal Sacrifice" could be change for: Sample Collection
- the authors applied any previous statistical normality test to know if the samples present or not a normal distribution. This test is very important and helpful to select a correct parametric or non-parametric test to evaluate the data. If yes, it is important insert the name of the test used in statistical description. If not, I recommend review the statical analyses.
Author Response
Comments from reviewer 1:
Comment 1: In the manuscript: Sex differences between neuronal loss and the early onset of amyloid deposits and behavioral consequences in 5xFAD transgenic mouse as a model for Alzheimer’s disease, the authors showed a significant gender influence in interesting parameters associated with neuronal loss and cognitive impairment in early stage of AD using a specific transgenic mice model with 4 or 6 age months and compared with WT. However, the authors cited in the introduction section it is known that 5xFAD mouse model present the AD related alterations in determinate ages: "The 5xFAD mouse model has been reported to encompass early-onset amyloid pathology including intraneuronal amyloid-beta (Aβ) and cognitive deficits at 1.5 months, plaque deposition in the subiculum and cortex deep layers at 2 months, spatial memory impairment at 4 – 5 months, and neuronal loss in the subiculum and cortical layer V at 9 months [8]'. On the other hand, the authors concluded that: "our study establishes the age of onset of multiple AD hallmark pathologies, including amyloid plaques, neuronal loss, learning deficits, spatial memory impairments, and hypoactivity. Through demonstrating onset age and gender differences". The question is if the amyloid plaques deposition and neuronal loss is report in the same animal model at 1.5 and 2 months, how the authors can conclude that they established the age of onset if they evaluated just 4- and 6-months age animals? The reported results could have been started early (e.g., 2 or 3 months)? In my opinion, a temporal study could confirm the information and also shows if male or female present that alterations at different ages or not. The authors should insert a shortly justificative to explain why they evaluated just 4- or 6-months age mice in the present study in introduction or discussion section.
Response 1: We would like to thank the reviewer for the detailed review of our manuscript. We understand the reviewer’s concern regarding the discrepancy in terms of pathology and behavioral deficits between the current study and previous ones. We agree with the reviewer and acknowledge that there is a need to clarify the age of animals we selected for analysis. While 5xFAD mice were indeed found to exhibit amyloid deposition and neuronal loss as early as 1.5-months old, they do not demonstrate cognitive impairment at that age. Our focus in the current study is to identify a valid parameter that can inform the emergence of behavioral deficit. While a temporal study on the molecular pathology of 5xFAD mice at earlier time points would be greatly beneficial to the field, analysis at a later time point where the mice have developed cognitive deficits aligns with the aim of the current study. We have included the justification of choosing mice aged 4-months and 6-months in the manuscript to emphasize our aim (Line 22-31; Line 47-48; Line 83-86).
Comment 2: Major review in Methods. In the Quantification of the chromogenic staining, it is not clarified the quantification criteria applied. The authors need describe more details.
- "Each region was delineated, and a threshold (half of the mean background) was applied in ImageJ (NIH, USA). The percentage area of positive staining with an intensity above the threshold was calculated" Is it means the results represent the integrated density of pixels?
- The quantification criteria were the same for all antibodies.
- How many sections of each animal was quantified for each marker?
- What was the interval in μm between sections?
- What criteria was applied for select a representative area?
- The quantification was performed in both right and left side of the brain? If not, why?
- To describe neuronal loss the authors performed an immunolabelling with NeuN marker and counted the number of positive cells. The authors consider both brain side. The count was performed in a representative area each side (fill image or a specific area).? The authors used a correction formula to obtain final result. (e.g., to corrects double counting of motoneurons, because the same cell may be present in two sections, it is commonly used the Abercrombie’s formula). Could be applicable for this brain neurons? In addition, is not clear how the NeuN results were converted to cell/mm2.
- How was performed the GFAP quantification? The data is showing as a relative expression but no explanation about that was find in methods.
Response 2: We would like to thank the reviewer for the effort of enhancing the quality of the current manuscript. To address the reviewer’s concerns, the quantification of 4G8 and GFAP staining is calculated by dividing the area with pixel intensity higher than the threshold by the total delineated area. The quantification criteria of 4G8 and GFAP staining are the same as both are pixel intensity-based, while that of NeuN staining was conducted manually by counting the number of NeuN+ neuronal nuclei in a selected area based on the outline delineating the subregion according to Mouse Brain Atlas by Franklin and Paxinos. The area of the selected region was calculated by ImageJ based on the scale bar in the photomicrograph. The number of positively-stained cells was counted and divided by the area of the selected region to produce the result in the unit of cell/mm2. Nevertheless, the criteria of a positively stained cell for all antibodies is the same – area with an intensity above the threshold. For each marker, 3-4 sections per animal were quantified. Sections were obtained from both right and left side of the brain and were subjected to randomization prior to quantification. During cryosection, 40 μm-thick brain sections were collected in a 5-series manner. Therefore, the interval between sections used for each antibody staining is approximately 200 μm. Given that the size of neuronal cell body is approximately 100 μm, staining conducted on brain sections collected in series manner here does not require correction for double counting. The photomicrograph showing the corresponding brain region that resulted in a value that is the closest to the mean value of the group was selected as the representative figure. We performed GFAP quantification in the same manner as 4G8 quantification, in which the area covered by the positively-stained cells was divided by the total area of the selected region. We acknowledge that ‘relative expression’ may not be an accurate description for GFAP results. We have changed that to ‘% GFAP positive area’, which, we believe, better describe the results. We have included all the necessary edits in the revised manuscript (Line 185-186).
Comment 3: Minor comments: Methods section:
- When we perform behavioral tests, it is very important detailed the time of occurred (e.g., day or night, always at the same hour or not), due the circadian cycle in rodents. Please include the complete information in the text.
- the item "Animal Sacrifice" could be change for: Sample Collection
- the authors applied any previous statistical normality test to know if the samples present or not a normal distribution. This test is very important and helpful to select a correct parametric or non-parametric test to evaluate the data. If yes, it is important insert the name of the test used in statistical description. If not, I recommend review the statical analyses.
Response 3: We appreciate the reviewer’s comments to strengthen the rigor of the current manuscript. We agree that the details of the behavioral tests conducted is essential. Hence, we have added additional description in Materials and Methods (Line 103-104) . We also agree with the reviewer that the term ‘sample collection’ is more appropriate and hence have amended as such (Line 150). We also added the requested information regarding statistical normality test in the statistical analysis section (Line 197-199).
Reviewer 2 Report
Poon and colleagues examine sex differences in pathology, neuronal loss and cognition in the 5xFAD mouse model of Alzheimer’s disease. The experiments are well conducted and analyzes and the manuscript is generally well written. While the findings are not groundbreaking, the results are valuable to the field as they contribute to a more thorough characterization of the stage and sex differences disease progression of an important AD mouse model. I generally support publication of the manuscript, but would suggest a number of additions and improvements prior to publication. My specific comments are as follows:
1) Though not a critical point, it seems odd to frame the study as in part looking for biomarkers that can aid early intervention in AD, when a main finding is that neuron loss may be a more accurate biomarker. Neuron loss as a ‘biomarker’ may be more predictive than amyloid, but it clearly is not a trait/biomarker that could allow for early intervention.
2) Line 305 and figure 2. The authors claim a reduction in neuron numbers in the text, but the figure shows a significant increase. Please clarify this discrepancy.
3) The authors examine GFAP-positive astrocytes in Fig. 4. It would be helpful if the authors provided GFAP cell counts as well as just GFAP ‘expression’, since GFAP signal can be increases by upregulation and/or proliferation.
4) Likewise, including an analysis of microgliosis and/or Iba1 staining to examine microglia in these same mouse cohorts would be a valuable addition.
5) The authors should include additional discussion of sex differences in amyloid, neuron loss and cognition in human AD patients, and how those compare to the findings in 5xFAD mice. Including more discussion of tau pathology in AD would also be appropriate.
6) The authors say on line 64 “intraneuronal amyloid-beta”. Is this a typo?
Author Response
Comment 1: Poon and colleagues examine sex differences in pathology, neuronal loss and cognition in the 5xFAD mouse model of Alzheimer’s disease. The experiments are well conducted and analyzes and the manuscript is generally well written. While the findings are not groundbreaking, the results are valuable to the field as they contribute to a more thorough characterization of the stage and sex differences disease progression of an important AD mouse model. I generally support publication of the manuscript, but would suggest a number of additions and improvements prior to publication. My specific comments are as follows:
Though not a critical point, it seems odd to frame the study as in part looking for biomarkers that can aid early intervention in AD, when a main finding is that neuron loss may be a more accurate biomarker. Neuron loss as a ‘biomarker’ may be more predictive than amyloid, but it clearly is not a trait/biomarker that could allow for early intervention.
Response 1: We would like to thank the reviewer for the helpful suggestion. After thorough discussion among the authors, we agree with the reviewer that biomarker for early intervention may not be the most appropriate term in describing the role of neuronal loss we have identified in the current study. The content of the revised manuscript was amended to better align with the aim of the current study
Comment 2: Line 305 and figure 2. The authors claim a reduction in neuron numbers in the text, but the figure shows a significant increase. Please clarify this discrepancy.
3) The authors examine GFAP-positive astrocytes in Fig. 4. It would be helpful if the authors provided GFAP cell counts as well as just GFAP ‘expression’, since GFAP signal can be increases by upregulation and/or proliferation.
4) Likewise, including an analysis of microgliosis and/or Iba1 staining to examine microglia in these same mouse cohorts would be a valuable addition.
5) The authors should include additional discussion of sex differences in amyloid, neuron loss and cognition in human AD patients, and how those compare to the findings in 5xFAD mice. Including more discussion of tau pathology in AD would also be appropriate.
6) The authors say on line 64 “intraneuronal amyloid-beta”. Is this a typo?
Response 2: We thank the reviewer for the in-depth examination of the manuscript. The ‘significant reduction in the neuronal number’ in section 3.3 was indeed a mistake. It has been corrected in the revised manuscript. For GFAP staining analysis, threshold analysis method was used for quantification, with the results presented in percentage of area covered by GFAP+ cells. While the total area covered by GFAP+ cells in the overall hippocampus represents the relative GFAP expression among different groups, the purpose of categorizing the analysis based on subregions in the hippocampus is to identify the region most significantly affected by neuroinflammation. We believe that threshold analysis method based on the staining conducted is sufficient to answer our current research question. While an additional Iba1 staining will be ideal and undoubtably strengthen the molecular characterization of 5xFAD mice, unfortunately we do not possess sufficient brain section samples for additional staining. As per the reviewer’s request, we have included additional discussion focusing on AD patients and the molecular disparity between male and female (Line 482-493). Regarding ‘intraneuronal amyloid-beta’, we referred to the amyloid species formed within neuronal cells prior to amyloid plaque deposition formed by the secreted amyloid-beta. We hope that this explanation clarifies the reviewer’s doubt.
Reviewer 3 Report
In this paper Poon and colleagues studied the correlation between neuronal loss, pathology and behavioural deficits in the 5xFAD mice at two different ages 4 and 6 months. There is a basic conceptual problem that is not addressed by the authors: how is it possible that the female 5xFAD have worst ab pathology and astrogliosis than males 5xFAD, similar neuronal loss and still perform better in all the behavioural tasks? What does it cause then the behavioural abnormality seen in males? Also, there are other discrepancies that need to be addressed:
· Fig. 1 E-F: These panels are too many and it is very hard to follow the results. The latency for males wt and 5xFAD should be plot in the same graph and same should be done for female. This is the best way to appreciate all the difference between genotypes and ages.
· Fig1 H, J: It is not clear from the results why it is important to do a short-term probe only 90s after the rehearsal learning.
· Fig. 2A: From the picture showed here it seems that in the subiculum area there are more plaques in 4-month-old mice compared to 6 months old males.
· Fig. 2A: The pictures are too small to really appreciate the difference in plaques density in most of the brain areas analysed. It would be more informative to do a Thioflavin staining instead.
· Fig. 2C-D: It seems that there is an increase in neuron density in the CA1 between 4 and 6 months 5xFAD males. How can the authors explain it?
· It is not clear from the methods how they analysed neuronal loss. Did they use stereology? Please describe the technique used to count neurons in more details.
· Fig. 3: The quantification graph does not represents what it is shown in the pictures. In fact, female wt at 4 months seem to have less neurons than the transgenic at the same age. Also, neuronal loss between 4 and 6 months 5xFAD males seems comparable instead of being more evident in 6 month old males like they showed in the quantification.
Author Response
Comment 1: In this paper Poon and colleagues studied the correlation between neuronal loss, pathology and behavioural deficits in the 5xFAD mice at two different ages 4 and 6 months. There is a basic conceptual problem that is not addressed by the authors: how is it possible that the female 5xFAD have worst ab pathology and astrogliosis than males 5xFAD, similar neuronal loss and still perform better in all the behavioural tasks? What does it cause then the behavioural abnormality seen in males? Also, there are other discrepancies that need to be addressed:
Fig. 1 E-F: These panels are too many and it is very hard to follow the results. The latency for males wt and 5xFAD should be plot in the same graph and same should be done for female. This is the best way to appreciate all the difference between genotypes and ages.
Fig1 H, J: It is not clear from the results why it is important to do a short-term probe only 90s after the rehearsal learning.
Response 1: We would like to thank the reviewer for the effort of increasing the readability of the current manuscript. We agree with the reviewer that it would be easier to comprehend the results when combined in the same graph. We have amended Figure 1 according to the reviewer’s comment (Line 289-292). The purpose of including both probe days is to assess whether the animals’ cognitive deficits were specific to short-term and/or long-term memory. The short-term probe was conducted 90 min after the last training session, which is a common protocol for examining short-term memory. From the results, we observed a decreased time spent in the target quadrant by 6-months old male and female 5xFAD mice, corroborating with data obtained from Y-maze forced alternation test that impaired short-term spatial memory begin to emerge at 6-months old in 5xFAD mice.
Comment 2: Fig. 2A: From the picture showed here it seems that in the subiculum area there are more plaques in 4-month-old mice compared to 6 months old males.
Fig. 2A: The pictures are too small to really appreciate the difference in plaques density in most of the brain areas analysed. It would be more informative to do a Thioflavin staining instead.
Fig. 2C-D: It seems that there is an increase in neuron density in the CA1 between 4 and 6 months 5xFAD males. How can the authors explain it?
It is not clear from the methods how they analysed neuronal loss. Did they use stereology? Please describe the technique used to count neurons in more details.
Response 2: We would like to thank the reviewer for the careful examination of the manuscript and figures. We understand the reviewer’s concern regarding the size of the representative figures shown. Hence, we have included an enlarged version of Fig. 2A in the revised manuscript for better visualization. Unfortunately we do not have sufficient brain sections for additional staining. Nevertheless, as 4G8 staining is a widely used and generally accepted approach for amyloid plaque quantification, we believe our current results are valid in terms of the significant difference of amyloid plaque deposition between 4-months and 6-months old 5xFAD mice. We would like to clarify what the reviewer may have mistaken in Fig. 2D. In the notations used to indicate pairs or groups with statistically significant difference, we used line with ticks to indicate specific groups that show difference, and smooth lines to include all groups underneath. Therefore, there is no statistically significant difference between 4-months and 6-months old male 5xFAD mice. For NeuN staining quantification, we have employed the threshold analysis method in combination with manual counting. NeuN+ neuronal nuclei was identified by ImageJ, which is determined by signals above threshold. Photomicrographs of the regions of interest with the same area were taken for further manual counting of NeuN+ cells.
Comment 3: Fig. 3: The quantification graph does not represents what it is shown in the pictures. In fact, female wt at 4 months seem to have less neurons than the transgenic at the same age. Also, neuronal loss between 4 and 6 months 5xFAD males seems comparable instead of being more evident in 6 month old males like they showed in the quantification.
Response 3: We would like to thank the reviewer for the detailed review of the current manuscript. After reviewing the manuscript, we agree with the reviewer that the representative figure in Fig. 3A indeed does not accurately reflect the data presented in Fig. 3B. Hence, we have chosen another photomicrograph to visualize the general trend of neuronal loss observed in 4 months old female 5xFAD mice. Moreover, the number of NeuN+ cells between 4 and 6 months-old male 5xFAD mice did not exhibit statistically significant discrepancy. Both the representative figures in Fig. 3A and the dot plot in Fig. 3B supported this observation.
Round 2
Reviewer 1 Report
Thank you for being accepted my suggestions.
I`m satisfied with the new version.
Congratulations!
Reviewer 3 Report
The authors have addressed in full all my concerns.